# Stereospecific Si-C coupling and remote control of axial chirality by enantioselective palladium-catalyzed hydrosilylation of maleimides

Xing-Wei Gu [1], Yu-Li Sun [1], Jia-Le Xie [1], Xing-Ben Wang [1], Zheng Xu [1], Guan-Wu Yin [1], Li Li [1], Ke-Fang Yang [1] & Li-Wen Xu [1,2] ✉

Hydrosilylation of unsaturated carbon-carbon bonds with hydrosilanes is a very important process to access organosilicon compounds and ranks as one of the most fundamental reactions in organic chemistry. However, catalytic asymmetric hydrosilylation of activated alkenes and internal alkenes has proven elusive, due to competing reduction of carbon-carbon double bond or isomerization processes. Herein, we report a highly enantioselective Si-C coupling by hydrosilylation of carbonyl-activated alkenes using a palladium catalyst with a chiral TADDOL-derived phosphoramidite ligand, which inhibits O-hydrosilylation/olefin reduction. The stereospecific Si-C coupling/hydrosilylation of maleimides affords a series of silyl succinimides with up to 99% yield, >99:1 diastereoselectivity and >99:1 enantioselectivity. The high degree of stereoselectivity exerts remote control of axial chirality, leading to functionalized, axially chiral succinimides which are versatile building blocks. The product utility is highlighted by the enantioselective construction of N-heterocycles bearing up to three stereocenters.

[1] Key Laboratory of Organosilicon Chemistry and Material Technology of Ministry of Education, and Key Laboratory of Organosilicon Material Technology of Zhejiang Province, Hangzhou Normal University, No. 2318, Yuhangtang Road, Hangzhou 311121, PR China. [2] State Key Laboratory for Oxo Synthesis and Selective Oxidation, Suzhou Research Institute (SRI) and Lanzhou Institute of Chemical Physics (LICP), University of Chinese Academy of Sciences (UCAS), Lanzhou 730000, PR China. ✉email: liwenxu@hznu.edu.cn

Silicon–carbon bond-forming reactions, including C–H/C–X silylation, hydrosilylation, and cross-exchange of Si–C bond, have been considered as the key topic and corner-stone of organosilicon chemistry and of great value in organic synthesis and functional materials[1–7]. However, the stereospecific construction of Si–C bond remains challenging and under-exploited, thereby preventing the enantioselective functionalisation of organosilicon compounds and downstream transformations. Here, we reported a highly enantioselective Si–C coupling hydrosilylation of carbonyl-activated alkenes using palladium catalysis with chiral TADDOL-derived phosphoramidite ligand, which inhibited previously common O-hydrosilylation/reduction of carbon–carbon double bond. This was proved in the enantioselective hydrosilylation in maleimides as well as the remote control of axial chirality of N-arylmaleimides via a single-step transformation. The products could be obtained in up to 99% yield, >99:1 diastereomeric ratio (for axial chirality) and >99:1 enantiomeric ratio. On the basis of experimental results, we elucidated the mechanistic details and the utility of the approach in synthetic chemistry and photocatalysis that was highlighted by the enantioselective construction of chiral N-heterocycles bearing one to three carbon-stereogenic centres.

Notably, hydrosilylation of unsaturated carbon–carbon bonds with hydrosilanes ranks one of the most fundamental reactions in industrial chemical production[8–14], such as the production of coupling silane and silicone rubber[15]. And recently it has become a very important process to access synthetically useful organosilicon compounds and chiral organosilanes[16–18] that are useful in asymmetric catalysis, functional materials, and can be employed as silicon-containing drug candidates. However, chiral silanes with functional groups are still difficult to be constructed via Si–C coupling due to the scarcity of highly enantioselective or broadly applicable methods. This limitation might discourage the pursuit of bioactive organosilicon compounds as drug candidates or chiral Si-based materials[19–21], despite functionalized silanes are exceptionally important for industrial processes or modern material technology across a wide range of disciplines. To date, asymmetric hydrosilylation is one of the core Si–C coupling transformations for the construction of chiral silanes. Very recently, several well-established protocols have been developed to the enantioselective hydrosilylation of terminal alkenes and alkynes for the synthesis of chiral organosilicon compounds with good regio- and enantioselectivities (Fig. 1a)[22–29], which has been recognised as a hot topic in organic synthesis in the past years. In contrast, the enantioselective hydrosilylation of internal alkenes or its analogues are uncommon, and the synthetic capabilities for the catalytic asymmetric hydrosilylation of activated alkenes are also presently limited[30,31]. Catalytic asymmetric hydrosilylation of activated alkenes and internal alkenes have been proven elusive, because the reduction of carbon–carbon double bond or isomerization is much less energetically favourable. For example, the hydrosilylation of EWG-activated alkenes may generate mixtures of reductive product, α- and β-adducts, and especially for α,β-unsaturated carbonyl compounds, silyl ketene acetals (O-silylation adduct with 1,4-addition), silyl ethers/amines (O-silylation adduct with 1,2-addition), and polymeric byproducts were also obtained depending on the catalyst systems (Fig. 1b)[32–34]. Although there have been much efforts made to achieve chemo- or regio-selective hydrosilylation of α,β-unsaturated carbonyl compounds, only a few examples of Si–C coupling C-silylation are known[35–37], and no successful report has been published on the enantioselective Si–C coupling hydrosilylation of α,β-unsaturated carbonyl compounds, to the best of our knowledge.

Because of the abundance and potentially bioactivity of imide/amide-containing nature products, maleimide and its derivatives are versatile building blocks for synthetic chemistry and functional

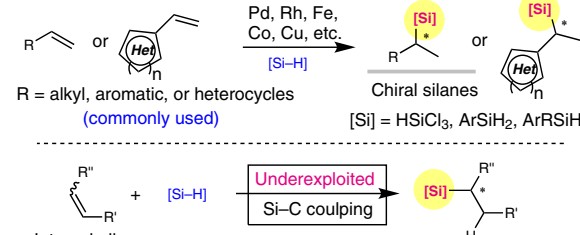

**a** Classic catalytic asymmetric hydrosilylation and its inherent challenges

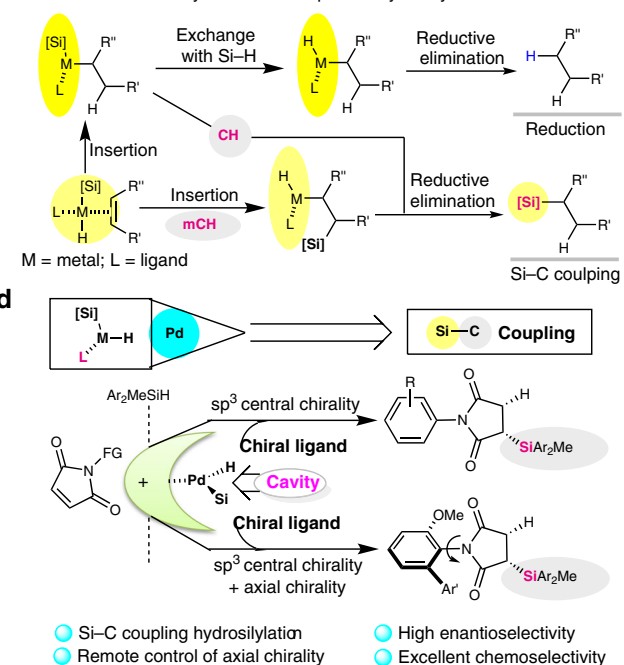

**b** Possible pathways in catalytic hydrsoilyaltion of carbonyl-activated alkenes

**c** The mechanistic analysis for the competitive hydrosilylation

M = metal; L = ligand

**d**

○ Si–C coupling hydrosilylation          ○ High enantioselectivity
○ Remote control of axial chirality      ○ Excellent chemoselectivity

**Fig. 1 Chemo- and stereo-selective issues with hydrosilylation. a** Catalytic asymmetric hydrosilylation of terminal alkenes mediated by transition-metal catalysts. **b** Traditional methods for hydrosilylation of EWG-activated alkenes led to reduction and O-silylation. **c** Two classic pathways for transition-metal-catalysed hydrosilylation of alkenes, and its mechanistic analysis should encourage greater adoption of Si–C coupling methods for the hydrosilylation of internal alkenes. **d** The catalytic desymmetric hydrosilylation of N-arylmaleimides for remote control of axial chirality. CH is Chalk–Harrod mechanism, and mCH is modified Chalk–Harrod mechanism.

materials[38–41], allowing for subsequent transformations of skeletal variation and investigation as drug candidates[42]. However, in contrast to the hydrosilylation of general electron-rich alkenes/alkynes, the synthetic power of hydrosilylation of electron-deficient alkenes, including maleimide, has not fully established[43,44] this stereoselectivity for Si–C coupling silylation of α,β-unsaturated carbonyl compounds has not been challenged. First, catalytic hydrosilylation of α,β-unsaturated carbonyl compounds usually

affords solely a conjugated reduction product, thus a catalyst system should be identified to overcome the expected reductive hydrosilylation. Second, the α,β-unsaturated carbonyl compounds containing functional carbonyl group, the catalyst must be capable of good group tolerance as well as effective control of enantioselectivity. In addition, for more than 30 years, especially after Hayashi's milestone discovery that Pd–MOP complexes catalysed Markovnikov-type asymmetric hydrosilylation with trichlorosilane[45, 46], the selectivity of palladium-catalysed Si–C coupling hydrosilylation of α,β-unsaturated carbonyl compounds has not been well established. Thus, this challenge motivates us to aim at developing an enantioselective hydrosilylation of maleimides, providing a straightforward approach to a wide range of silyl-functionalized carbonyl compounds that can enrich the chiral synthesis toolbox with distinct chemo- and stereo-seleletivity. Based on the analysis of reaction mechanisms for metal-catalysed hydrosilylation and reduction with hydrosilanes revealing the potential migratory insertion and ligand-controlled hydride transfer (Fig. 1c), and inspired by previous reports on the role of secondary interactions in the asymmetric palladium-catalysed hydrosilylation of olefins with chiral monophosphine ligands[47], we envisioned that an appropriate ligand with a cavity-like structure could reverse the chemoselectivity from reduction to Si–C coupling silylation and might control the enantioselectivity of hydrosilane addition to C=C bond of maleimides. Therefore, if a chiral ligand bearing a bulky and cavity-like group as well as displaying suitable secondary interactions could be beneficial to the formation of a proton shuttle for subsequent Si–C coupling hydrosilylation, a highly enantioselective hydrosilylation of EWG-activated alkenes (EWG: electron-withdrawing group) would be achieved.

Herein, we report our recent efforts to establish a palladium-catalysed protocol for the catalytic asymmetric hydrosilylation of maleimides with hydrosilanes to provide silylated carbon stereocenters, with high chemo- and enantioselectivity (Fig. 1d). Through the asymmetric palladium-catalysed hydrosilylation of N-arylmaleimides, the remote control of axial chirality of atropisomeric succinimides could be achieved during the Si–C coupling. The key feature of present methodology is the ability of the palladium catalyst to exert stereochemical induction from functionalisation of C–C double bond to the formation of the remote C–N axis[48, 49] via a single Si–C bond-forming process unlike previously reported Michael addition[50–53] or well-established cycloaddition[54–57]. Computational studies reveal the mechanism of the palladium-catalysed C-silylation and support the unusual stereoselectivity.

## Results

**Optimisation of reaction conditions**. We first carried out a model hydrosilylation involving N-phenylmaleimide **1a** and the commercially available diphenylmethylsilane **2a** with chiral P-ligand and Pd$_2$(dba)$_3$·CHCl$_3$. Unfortunately, owing to competition of reduction and Si–C coupling hydrosilylation, most of commercially available phosphine ligands, such as 2,2′-bis(diphenylphosphino)-1,1′-binaphthyl (BINAP), 2-(diphenylphosphino)-2′-methoxy-1,1′-binaphthyl (MOP), and other P-ligands resulted into reductive products but only with trace amount of desired products (Entries 14–18 of Table 1). And as expected, it was found that only chiral TADDOL-derived phosphoramidites bearing aromatic bulky groups could give low to moderate yields of desired silyl product **3a** in the palladium-catalysed hydrosilylation of N-phenylmaleimide (Table 1 and Supplementary Table 1, TADDOL = 1,1,4,4-tetra-aryl-2,3-O-isopropylidene-L-threitol). In the event, at 60 °C and after 18 h the conversion of **1a** was completely and the desired product was obtained in 68% with

94% ee in the presence of chiral TADDOL-derived phosphoramidite **L12** (Entry 12 of Table 1). These experimental data showed the steric repulsion of chiral P-ligand inhibit the reductive O-hydrosilylation. However, small amounts of the side product (32% 1-phenyl-pyrrolidine-2,5-dione) were still detected because of reductive hydrosilylation. The formation of reductive product implies that the pathway involving Si–C coupling is accompanied by Si–H activation and subsequent hydrogen-transfer to Pd–Si bond (Fig. 1c). Under the same conditions but with other TADDOL-derived phosphoramidites resulted into decreased chemoselectivities (from 4:96 to 59:41 c.r.) and enantioselectivities (31–93% ee, see entries 1–14 of Table 1). Next, we examined the effect of palladium catalyst precursors (Supplementary Table 2). In certain cases better chemoselectivity was generated with the same good enantioselectivity when Pd$_2$(dba)$_3$ as Pd catalyst (85% yield of **3a**, 94% ee). Then after an extensive evaluation of solvents, reaction temperature, and phosphorous ligands, the Pd$_2$(dba)$_3$/**L12** was identified as an effective catalyst for the model hydrosilylation of N-arylmaleimide in toluene at 50 °C (entry 20 of Table 1, for **3a**, 93:7 c.r., 96% ee). Notably, some of experimental data were unexpected, for example, higher temperature was beneficial to the Si–C coupling hydrosilylation but not the reduction (Supplementary Table 4). However, the reaction performed at higher temperature was sacrificed in term of stereoselectivity to some extent.

**Scope of the palladium-catalysed hydrosilylation of maleimides**. We then evaluated a series of maleimides to probe the reaction scope to generate the synthetic information about the stereospecific Si–C coupling hydrosilylation (Fig. 2b). Various maleimides with an aryl unit (**1a–1w**), whether it is electron-withdrawing or electron-donating, react efficiently with hydrosilane **2a** to give desired products (**3a–3w**) in excellent enantioselectivities (93–99% ees) and moderate to good yields (up to 99% yield). It should be noted that the coordination capacity of the Pd–L complex with the activated alkene substrate possibly decreases due to the weakened solubility in the toluene and unexpected electronic properties of maleimides that with OR groups, resulting into a decreased yield of corresponding products, such as the representative examples of **3d**, **3e**, and **3g**. In addition, it was observed in experiments that the standard silica gel chromatography purification of the product led to the formation of succinimides 4 by silica gel promoted desilylation of the products. More specially, N-unsubstituted maleimide **1x** is also tolerated, which is notable since a NH group can be proven to be no interference during undergoing Si–C coupling hydrosilylation, albeit the enantioselectivity is slightly decreased as 81% ee with 69% yield. In addition, high efficiency and stereoselectivity was observed with N-alkylmaleimides (**1y–1bb**). A substrate bearing an additional N- or S-heterocycle performed smoothly under the optimised reaction conditions to deliver the desired product (**3cc** or **3dd** with 84% yield and 96% ee).

We subsequently examined the possibility of the catalytic asymmetric hydrosilylation for the remote control of axial chirality when the N-arylmaleimide substrates bearing bulky group at ortho-position of aryl unit were used (Fig. 2c). The influence of bulky substitution on the aryl ring was investigated to probe the steric effect and it was found that the tert-butyl group with large B values (about 15.5)[58] guaranteed the perfect and remote control of axial chirality (for **3gg**, 99% ee with 99:1 d.r.). Notably, except **3gg**, the atropisomers of mono-substituted N-arylmaleimide-derived silyl products **3ee–3hh** were co-existed in this protocol, which difficultly affords the pure and single atropisomer in the palladium-catalysed hydrosilylation.

**Table 1 Optimisation of reaction conditions[a].**

| Entry | Ligand | Conversion (%)[b] | 3a/4a[b] | ee% of 3a[c] | Entry | Ligand | Conversion (%)[b] | 3a/4a[b] | ee% of 3a[c] |
|---|---|---|---|---|---|---|---|---|---|
| 1 | L1 | 97 | 25:75 | 70 | 14 | L14 | 14 | <1:99 | ND |
| 2 | L2 | 95 | 17:83 | 69 | 15 | L15 | >99 | <1:99 | ND |
| 3 | L3 | >99 | 30:70 | 68 | 16 | L16 | 28 | <1:99 | ND |
| 4 | L4 | >99 | 29:71 | 76 | 17 | L17 | 29 | <1:99 | ND |
| 5 | L5 | trace | ND | ND | 18 | L18 | >99 | <1:99 | ND |
| 6 | L6 | >99 | 8:92 | 62 | **Entry** | **Solvent[d]** | **Conversion (%)** | **3a/4a[b]** | **ee% of 3a[c]** |
| 7 | L7 | >99 | 59:41 | 79 | 19 | Dioxane | 99 | 67:32 | 92 |
| 8 | L8 | 93 | 50:50 | 79 | 20[e] | Toluene | >99 | 93:7 | 96 |
| 9 | L9 | >99 | 41:59 | 93 | 21 | Et$_2$O | >99 | 33:67 | 92 |
| 10 | L10 | 90 | 4.5:95.5 | 91 | 22 | DCE | >99 | 85:15 | 94 |
| 11 | L11 | >99 | 4:96 | 86 | 23 | THF | >99 | 4:96 | 70 |
| 12 | L12 | >99 | 68:32 | 94 | 24 | DCM | >99 | 96:4 | 93 |
| 13 | L13 | 95 | 4:96 | 31 | | | | | |

The structure of chiral ligands in this work:

L6, Ar = 4-OMe-Ph
L7, Ar = 4-CF$_3$-Ph
L8, Ar = 3-CF$_3$-Ph
L9, Ar = 3,5-CF$_3$-Ph

L1, Ar = 4-Et-Ph
L2, Ar = 4-$^i$Pr-Ph
L3, Ar = 4-$^t$Bu-Ph
L4, Ar = 4-TMS-Ph
L5, Ar = 4-F-3,5-TMS-Ph

L10, X = Ph
L11, X = NEt$_2$
L12, X = Pyrrolidineyl
L13, Ar = C$_6$F$_5$
L19, Ar = 4-CF$_3$-Ph

L14, R = PPh$_2$
L15, R = OMe

L16

L17,(R)-DTBM-SEGPHOS
Ar = 3,5-(t-Bu)$_2$-4-MeO-C$_6$H$_2$

L18

[a]All the reactions were run on a 0.1 mmol scale in 1.0 mL DCE (entries 1–13) or toluene (entries 14–18) at 60 °C for 18 h.
[b]Determined by 1H NMR using dibromomethane as an internal standard.
[c]Determined by chiral HPLC.
[d]The reactions were performed with Pd$_2$(dba)$_3$ and ligand L12.
[e]The reaction temperature is 50 °C.

**Remote control of axial chirality by palladium-catalysed hydrosilylation.** To construct axially chiral *N*-arylmaleimide derivatives via catalytic asymmetric hydrosilylation, we envisioned the introduction of two groups on the *ortho*-position of *N*-arylmaleimides to stabilise the atropisomeric chirality during the remote control by palladium catalyst (Fig. 3a). Steric modification of one of the substituents on **3hh** by an aryl group could counter racemisation at high temperature. Owing to unexpected effect of much bulky group on catalytic asymmetric hydrosilylation, the other concern was also that reductive side-products could form. Fortunately, no detrimental effects to the chemo- and stereo-selectivity of this process were observed when biarylmaleimides were used (Fig. 3b), and all the atropisomeric succinimides **5a–5z** bearing an additional *sp*$^3$-central chirality that with a high energy barrier for C–N bond rotation (for example, the energy barrier for compound **5a** is 44.2 kcal/mol, see Supplementary Fig. 10) were obtained in moderate to high yields (up to 97% yield) and good

enantioselectivities (up to 95% *ee*) as well as excellent diastereoselectivity (up to >99:1 d.r.). An assortment of easily available *N*-arylmaleimides was viable for this stereospecific Si–C coupling hydrosilylation as well as remote control of axial chirality of C–N bond. Expanding the substrate scope to investigate hydrosilanes permitted us to forge other arylsilanes for the catalytic asymmetric hydrosilylation. For example, the atropisomeric succinimides **5v–5y** could be also achieved in excellent diastereo- and enantioselectivity (91-95% *ee* and 97:3 to >99:1 d.r.). However, bulky arylsilane that containing *t*-Bu group on aromatic ring slowly reacted to form the desired atropisomeric succinimide **5w** under the same reaction conditions, which revealed the steric repulsion between arylsilane and *ortho*-substituted *N*-arylmaleimide is not inconsiderable. Lastly, a benzofuran-containing *N*-arylmaleimide was also successfully Si–C coupling to generate the desired silyl atropisomer in 87% yield and 95% *ee*. Notably, the absolute configuration of the atropisomeric succinimide (*P,S*)-**5f**

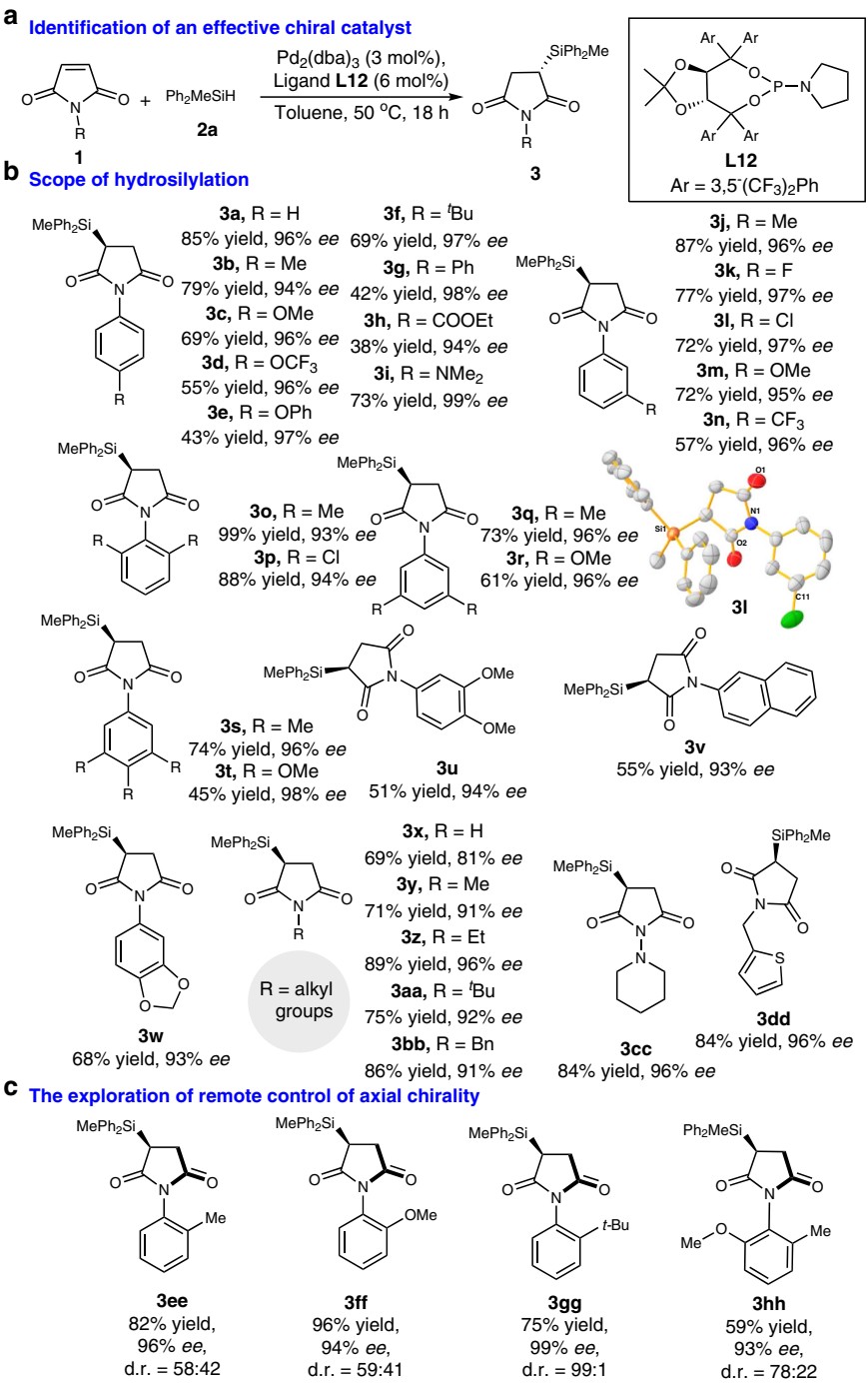

**Fig. 2 Enantioselective palladium-catalysed hydrosilylation of maleimides. a** The determination of chiral palladium catalyst after screening of a variety of chiral ligands and reaction parameters, and that corresponding to **L12** is optimal. **b** The catalytic asymmetric hydrosilylation of maleimides is broadly applicable, affording silyl products with excellent enantioselectivities (up to 99% ee). Products containing S-heterocycle and unsubstituted imide can be accessed. **c** The effect of bulky groups at the *ortho*-position of *N*-arylmaleimides on enantioselective Si–C coupling hydrosilylation with respect to the remote control of axial chirality.

was suitable for the determination of the remote control of axial chirality as *P* through X-ray crystallography analysis.

**Downstream transformations of enantiomerically enriched silyl succinimides**. To demonstrate the practicability of the catalytic asymmetric hydrosilylation, we carried out gram-scale reaction for the model transformation of **1a** (Fig. 4a), which offered the same high enantiomeric excess (ee) compared with the

small-scale process. Then, the enantiomerically enriched silyl succinimides can be converted to pyrrolidine and its derivatives, and the silyl group could be acted as a removable placeholder or masked auxiliary in these transformations. Synthesis of chiral *N*-aryl amino alcohol **7** under the Fleming–Tamao's oxidation conditions[59] showed the Si-linked *sp*^3 central chirality can be retained with high ee value (96% ee). This transformation offers an attractive entry for synthesis of chiral amino alcohols that

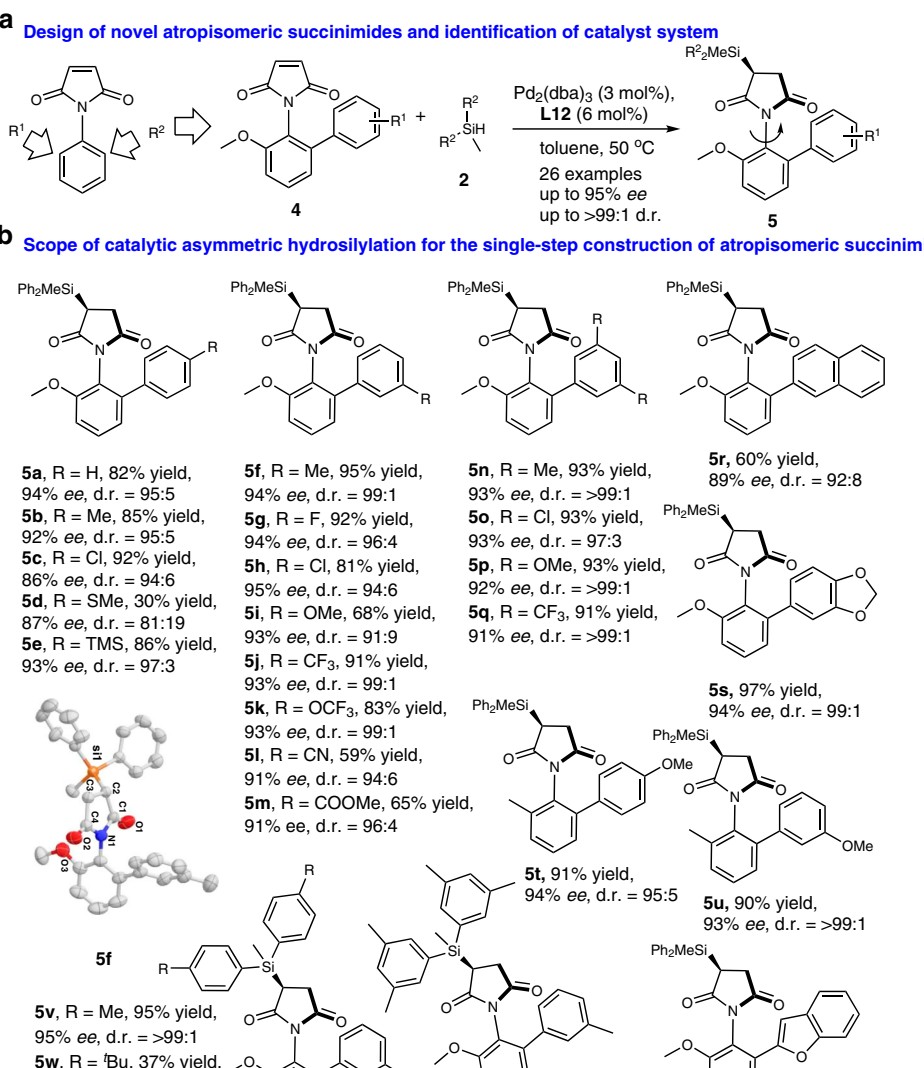

**Fig. 3 Remote control of axial chirality of C–N bond by hydrosilylation. a** The design of C–N bond rotatable atropisomers by introduction of two different substituents on the *ortho*-position of *N*-arylmaleimides to stabilise axial chirality. **b** The catalytic asymmetric hydrosilylation is broadly applicable in the construction of atropisomeric *N*-arylmaleimides with good chemo-, diastereo-, and enantioselectivity and in up to 97% yield. Products containing *sp*³ central chirality and axial chirality can be accessed in the single step of Si–C coupling hydrosilylation.

could be used as precursors to biologically active molecules. In contrast, the oxidation of silyl succinimides **3** under the same reaction conditions, the desilylation product, 1-phenyl-pyrrolidine-2,5-dione, was formed exclusively. Recent progress on the radical cation-induced cyclisation of simple *N*-arylpyrrolidine with iodonium ylides leading to a broad range of indoline derivatives[60] inspired us to evaluate the placeholder effect of silyl group. And as already mentioned in this work, there is no catalytic asymmetric version for the enantioselective synthesis of indoline derivatives. Under the reported reaction conditions, the corresponding product **8**, containing an additionally carbon-stereogenic centre, was obtained in good enantioselectivity and promising diastereoselectivity (Fig. 4b). Unexpectedly, the more hindered *ortho*-position to the silyl group and amino group was the preferential cyclisation site, which is different from that of generally accepted pathway. The site-selective functionalisation was ascribed to the β-effect of silicon that featured with the

hyperconjugation stabilisation of the cation intermediate by the Si–C σ-bond[61]. Recent reports revealed the powerful potential of photoredox catalysis in the functionalisation of C–H bond[62–65], which promote us to evaluate the chirality transfer of Si-linked carbon stereocenter on chiral silyl arylpyrrolidine to stereoselective photocatalysed C–H functionalisation. We believed the chiral silyl group as a masked hydrogen or hydroxyl group plays important role in the silicon-mediated construction of carbon-stereogenic centre during photoredox-catalysed C–H functionalisation, providing supplementary method to build chiral molecules that it is a difficult and challenging task in photocatalytic reaction system[66]. Considered the importance of photocatalysed dual C–H bond functionalisation (dehydrogenation/[2 + 2] cycloaddition) of pyrrolidine that reported by Xu et al.[67], we checked the asymmetric version with chiral *N*-arylpyrrolidine **6** as starting material under the standard reaction conditions. The synthetic potential of this process can be supported by the perfect

**Fig. 4 Gram-sale synthesis and functionalisation and demonstration of utility. a** Gram-scale reaction. Compared with the small-scale reaction, using substrates **1a** and **2a** in a gram-scale reaction produced the desired product **3a** with the same level of enantiomeric excess (*ee*) and yield. **b** The silyl succinimides can be readily reduced to the stable silylpyrrolidine that could be transferred into structurally diverse and chiral *N*-heterocycles without racemisation. DMAD dimethyl acetylenedicarboxylate, PFNB petafluoronitrobenzene, CFL compact fluorescent light, ct the value of chirality transfer.

chirality transfer (100% *ct*) that product **9** was obtained in >99:1 *d.r.* and 96% *ee*. Therefore, these experiments in Fig. 4b suggested that one emerging application of placeholder effect of silyl group is its use in the enantioselective functionalisation of *sp*³ C–H bonds via silicon-mediated chirality transfer.

## Discussion
To clarify the stereospecific Si–C coupling process by palladium-catalysed hydrosilylation of maleimides, DFT calculations revealed that high chemo- and enantioselectivity originates from the aromatic interaction and steric repulsion caused by chiral TADDOL-derived phosphoramidite acting with the Pd–Si intermediate's aryl unit, in turn coordinated with carbon–carbon double bond of maleimide. In the more favourable pathway (path A of Supplementary Fig. 8) migratory insertion of an Pd/olefin hydride complex followed by reductive Si–C coupling (simplified as Chalk–Harrod mechanism)[68, 69]. Thus the Si–C coupling hydrosilylation but not olefin reduction occurred due to the irreversible β-hydride elimination that inhibited by the steric repulsion of chiral ligand. And the control experiments with deuterium labelling studies (KIE) determined the Si–H action is a key step in this reaction (Supplementary Fig. 3), indicating that

crucial role of bulky P-ligand bearing a large cavity in the directing migratory insertion of hydride to carbon–carbon double bond. The formation of H–Pd–Si intermediate within suitable cavity-type structure as a chiral proton shuttle arose from a TADDOL-derived phosphoramidite is necessary for the high enantioselectivity during the stereospecific migratory insertion of an olefin hydride complex, which could be supported by the ³¹P NMR analysis, nonlinear effect, and DFT calculations (Supplementary Figs. 7, 8). In addition, the absolute configuration of the silyl products **3** or **5** mainly depends on the cavity size of the ligand, which is determined by the extension direction of the four aryl groups on TADDOL-derived phosphoramidite **L12**, especially two of them. And one of the aryl groups that regulated the steric hindrance is perpendicular to the maleimide substrate, in which a chiral wall as an external cavity surface can be formed to control the stereoselective direction of hydrosilylation of maleimide that is conducive to highly enantioselective addition via the option of less steric repulsion.

In summary, the stereospecific Si–C coupling hydrosilylation of maleimides affords a series of silyl succinimides with the aid of stable and reactive Pd catalyst. Through the development of Pd-catalysed hydrosilylation, we achieved remote hydrosilylation

–controlled construction of C–N axial chirality within *ortho*-substituted *N*-arylmaleimides. Owing to the compatibility of steric repulsion as well as electron-rich and electron-deficient substituents, this Si–C coupling hydrosilylation reaction provides a way to diversify synthetically useful intermediates and complex molecules benefited from the concept of silicon-mediated organic synthesis.

## Methods

**General procedure for the palladium-catalysed hydrosilylation**. A vial was charged with *N*-arylmaleimide **1** (0.3 mmol), Pd$_2$(dba)$_3$ (8.2 mg, 3.0 mol%), (*R*,*R*)-**L12** (20.1 mg, 6 mol%), and evacuated under high vacuum and backfilled with N$_2$. Toluene (3 mL) was added subsequently. The mixture was stirred at 25 °C for 10 min, then the Ph$_2$MeSiH (0.6 mmol) was added to the reaction. The mixture was stirred at 50 °C in a preheated oil. Upon reaction completion, the mixture was filtered over a plug of silica gel (washed with 50 ml EtOAc), and the filtrate was concentrated. The crude was purified by column chromatography to give the corresponding product.

Full experimental details and the characterisation of compounds **3–8** are provided in the Supplementary Information.

## Data availability

The X-ray crystallographic coordinates for structures reported in this study have been deposited at the Cambridge Crystallographic Data Centre (CCDC), under deposition numbers 1967248 and 1994220. These data can be obtained free of charge from The Cambridge Crystallographic Data Centre via www.ccdc.cam.ac.uk/data_request/cif. All the data generated and analysed in this study, including the experimental details, spectra for all unknown compounds, and computational modelling data associated with all of the tables and figures, see Supplementary Files. All data underlying the findings of this work are available from the corresponding author upon reasonable request. The source data underlying Supplementary Figs. 5–8 and 11 are provided as a Source Data file. Source data are provided with this paper.

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

## Acknowledgements

This work was supported by the grants of National Natural Science Foundation of China (NSFC Nos. 21773051, 21703051, and 21901056), and Zhejiang Provincial Natural Science Foundation of China (LZ18B020001, LQ 19B040001, and LY18B020013). The authors thank K.Z. Jiang and X.Q. Xiao for their assistance on the MS and X-ray crystallographic analysis.

## Author contributions

L.-W.X. conceived the concept. X.-W.G., Y.-L.S, J.-L.X., and X.-B.W carried out experiments, including the preparation of chiral ligands and the palladium-catalysed hydrosilylation. Y.-L.S. repeated the reaction results and carried out the X-ray crystallographic analysis of **5f**. X.-W.G. and Z.X. carried out the DFT calculations. G.-W.Y., L.L., and K.-F.Y. synthesised the substrates and conducted the structural analysis of new compounds. L.-W.X wrote the manuscript and all authors discussed the results and participated in revising the manuscript. L.-W.X. supervised the project.

## Competing interests

The authors declare no competing interests.
