## [Peer Review File · Nature Communications]

Reviewers' comments:

Reviewer #1 (Remarks to the Author):

The manuscript by Xu and collaborators is an interesting example of a Pd catalyzed hydrosilylation of activated maleimides. The synthesis of novel organosilicon compounds is realized in an enantioselective fashion. The optimization is well performed and allows to find the best conditions to obtain a very high enantiocontrol. A limitation is given by the yields which are not always very high and do not reflect the same efficiency exerted by the catalyst on the enantioselectivity. In general, the scope of the hydrosilylation is quite good and a large number of alkyl and aryl maleimides is employed. The scope is furthermore implemented by the use of arylmaleimides with ortho substituents which drives the reaction to an enantio- atropo- and diastereoselective hydrosilylation process. The manuscript surely reports a novel reaction which is however limited only to maleimides as activated alkenes. However, the lack of generality is compensated with a rare example of axially chiral organosilicon compounds. Despite the concept of the desymmetrization of prochiral maleimides is not new, the novelty can be found in the way the desymmetrization is realized that is by means the Pd catalyzed hydrosilylation. The manuscript is of good quality but in my personal opinion sufficient for publication on Nature communication after the following revisions:

- 1) Novel axially chiral compounds have been prepared. However, no full characterization has been provided. It is necessary for some representative examples to have the experimental value of the rotational barrier in order to know the stability of this new kind of products.
- 2) The absolute configuration has been determined via X-ray analysis on compound 3. On compounds 5 the absolute configuration has been obtained through experimental ECD compared with the DFT simulated CD of a certain enantiomer. Which criteria has been followed by the author to use for the simulation the reported diastereoisomer? Are there any experimental evidences for this. In fact, It appears strange that the silicon group is syn to the largest ortho substituent of the N-Aryl moiety. The B-value for the Ph group is 7.5 and for the OMe group is 5.6 see: Chem. Eur. J. 2009, 2645. Indeed, as reported in other examples of atroposelective desymmetrization, the addition to the maleimide double bond is anti to the large substituent of the aryl group: Tetrahedron 2016, 72, 5191, Org. Lett. 2015, 17, 1728, Synthesis, 2017, 49, 1519, Tetrahedron 2007, 8529. The correlation between the configuration of the chiral center in compounds 3 and 5 is too risky and it is necessary to determine the relative configuration of the 5 series and then their absolute configuration and eventually re-discuss the mechanism of the reaction.
- 4) In the case of compounds 3, yields are not always at high level. Is there a plausible explanation for this poor result. In the case of compounds 3d, 3e, 3g as representative examples, do you have traces of the corresponding succinimides 4?
- 3) Together with the previous references, the following should be added: Chem. Rev. 2015, 115, 11239

Reviewer #2 (Remarks to the Author):

Xu and co-workers reported a highly enantioselective hydrosilylation of maleimides using palladium catalysis, in which the stereospecific preparation of a series of silyl succinimides was achieved in up to 99% yield, and >99:1 er. And the remote control of axial chirality for efficient synthesis of functionalized succinimides bearing axial C-N is also very interesting. This is very good and thorough work, and the novel aspects in the stereoselective synthesis of these point and axially chiral silylated maleimides are well recognized. The reaction could be easily scaled up and the products could be also derivatized. This reviewer would recommend this manuscript for publication on Nature Communications after minor revision.

- (1) Figure 1 looks a bit messy, it is recommended to repaint it more beautifully. In fig.1b, this challenge is not that strong. Because in the computational studies, the hydride was added first which is same as classic Michael addition. It should be noted that the hydrosilylation of

unsaturated compounds to access α -silylated products has been well established. In fig. 1c, the "OA" should be "Insertion".

(2) For the Line 145, the "a Si-H activation-driven palladium complex shuttle" could be replaced by "palladium catalysis". Because the palladium complexes have not been well studied.

(3) A recent review on hydrofunctionalization of alkenes including hydrosilylation should be cited, such as *Chin. J. Chem.* 2018, 36, 1075–1109. A recent paper on asymmetric hydrosilylation of vinylsilanes should be cited, such as *Chin. J. Chem.* 2019, 37, 457–461.

(4) It will be better to attach the F NMR spectra for the fluorine-containing compounds.

(5) The English grammar should be corrected for some expressions, for example, there are a little cumbersome in Introduction part, please carefully modify them.

Dear Referees,

Thank you very much for your positive comments regarding our above manuscript. We have read your comments carefully and made correction accordingly. In this revised version, we highlighted the changes made by giving the text a yellow background. Specifically, the following revisions were made:

Revisions made in reply to referee 1' comments:

Comments:

The manuscript by Xu and collaborators is an interesting example of a Pd catalyzed hydrosilylation of activated maleimides. The synthesis of novel organosilicon compounds is realized in an enantioselective fashion. The optimization is well performed and allows to find the best conditions to obtain a very high enantiocontrol. A limitation is given by the yields which are not always very high and do not reflect the same efficiency exerted by the catalyst on the enantioselectivity. In general, the scope of the hydrosilylation is quite good and a large number of alkyl and aryl maleimides is employed. The scope is furthermore implemented by the use of arylmaleimides with ortho substituents which drives the reaction to an enantio-atropo- and diastereoselective hydrosilylation process. The manuscript surely reports a novel reaction which is however limited only to maleimides as activated alkenes. However, the lack of generality is compensated with a rare example of axially chiral organosilicon compounds. Despite the concept of the desymmetrization of prochiral maleimides is not new, the novelty can be found in the way the desymmetrization is realized that is by means the Pd catalyzed hydrosilylation. The manuscript is of good quality but in my personal opinion sufficient for publication on Nature communication after the following revisions:

1) Novel axially chiral compounds have been prepared. However, no full characterization has been provided. It is necessary for some representative examples to have the experimental value of the rotational barrier in order to know the stability of this new kind of products.

■ Response: Thank the referee for his suggestion. We have gotten the data for the rotation

barrier for some representative examples. It was found that the desired axial chiral products 5 possessed a high energy barrier for C-N bond rotation (>30 kcal/mol) Accordingly, we have added these experiment results in the Supplementary Material (Figure S16), in which these results revealed the stability of these axial chiral products shown in Figure 3.

2) The absolute configuration has been determined via X-ray analysis on compound 3. On compounds 5 the absolute configuration has been obtained through experimental ECD compared with the DFT simulated CD of a certain enantiomer. Which criteria has been followed by the author to use for the simulation the reported diastereoisomer? Are there any experimental evidences for this. In fact, It appears strange that the silicon group is syn to the largest ortho substituent of the N-Aryl moiety. The B-value for the Ph group is 7.5 and for the OMe group is 5.6 see: Chem. Eur. J. 2009, 2645. Indeed, as reported in other examples of atroposelective desymmetrization, the addition to the maleimide double bond is anti to the large substituent of the aryl group: Tetrahedron 2016, 72, 5191, Org. Lett. 2015, 17, 1728, Synthesis, 2017, 49, 1519, Tetrahedron 2007, 8529. The correlation between the configuration of the chiral center in compounds 3 and 5 is too risky and it is necessary to determine the relative configuration of the 5 series and then their absolute configuration and eventually re-discuss the mechanism of the reaction.

■ Response: Thank the referee for his useful information and related literatures. We have carefully studied these reports and found it is different largely from previous findings. For example, for the “Organocatalytic Atroposelective Formal Diels–Alder Desymmetrization of N-Aryl maleimides” (Org. Lett. 2015, 17, 1728) and similar organocatalysis (Tetrahedron 2016, 72, 5191 and Synlett 2017, 49, 1519), the presence of the tert-butyl substituent has a deep impact on the tandem enamine–iminium mechanism of the reaction highlighting the stereochemical outcome observed depends on the endo way that the α,β -unsaturated enamine and the maleimide approach each other. And for other cases, different ligands with changed mode of activation would give possibly opposite configuration (see Tetrahedron 2007, 8529). In our case, under same reaction conditions, including the same P-ligand, it was found different groups on ortho-substitution can't change the direction for the hydrosilylation process, thus the absolute configuration of carbon-stereogenic center could be confirmed by

product 3. And then on the basis of our DFT calculation on reaction mechanism and the calculated/experimental results on CD, we believed the confirmation of absolute configuration of product 5 is reasonable. Finally, we have added these references for related descriptions in the revised version.

3) Together with the previous references, the following should be added: Chem. Rev. 2015, 115, 11239

■ Response: We have added these references in the revised version.

4) In the case of compounds 3, yields are not always at high level. Is there a plausible explanation for this poor result. In the case of compounds 3d, 3e, 3g as representative examples, do you have traces of the corresponding succinimides 4?

■ Response: Thank the referee for his careful check. As shown in Figure 2, the coordination capacity of the Pd-L complex with the activated alkene substrate possibly decreases due to the weakened solubility in the toluene and unexpected electronic properties of maleimides that with OR groups, resulting into a decreased yield of corresponding products, such as the representative examples of 3d, 3e, and 3g. In addition, it was observed in experiments that the standard silica gel chromatography purification of the product led to the formation of succinimides 4 by silica gel promoted desilylation of the products. Accordingly, we have added these descriptions in the revised version (page 11, the last paragraph and the continued paragraph of page 13).

Revisions made in reply to referee 2' comments:

Comments:

Xu and co-workers reported a highly enantioselective hydrosilylation of maleimides using palladium catalysis, in which the stereospecific preparation of a series of silyl succinimides was achieved in up to 99% yield, and >99:1 er. And the remote control of axial chirality for efficient synthesis of functionalized succinimides bearing axial C-N is also very interesting. This is very good and thorough work, and the novel aspects in the stereoselective synthesis of these point and axially chiral silylated maleimides are well recognized. The reaction could be

easily scaled up and the products could be also derivatized. This reviewer would recommend this manuscript for publication on Nature Communications after minor revision.

(1) Figure 1 looks a bit messy, it is recommended to repaint it more beautifully. In fig.1b, this challenge is not that strong. Because in the computational studies, the hydride was added first which is same as classic Michael addition. It should be noted that the hydrosilylation of unsaturated compounds to access α -silylated products has been well establish. In fig. 1c, the “OA” should be “Insertion”.

■ Response: Thank the referee for his careful check, we have corrected these typos accordingly, for example, the “Challenges” is corrected as “Possible pathways” in Figure 1b.

(2) For the Line 145, the “a Si-H activation-driven palladium complex shuttle” could be replaced by “palladium catalysis”. Because the palladium complexes have not been well studied.

■ Response: Thank the referee for his careful check, we have corrected it accordingly.

(3) A recent review on hydrofunctionalization of alkenes including hydrosilylation should be cited, such as Chin. J. Chem. 2018, 36, 1075–1109. A recent paper on asymmetric hydrosilylation of vinylsilanes should be cited, such as Chin. J. Chem. 2019, 37, 457–461.

■ Response: Thank the referee for his suggestion, we have added these references in the revised version.

(4) It will be better to attach the F NMR spectra for the fluorine-containing compounds.

■ Response: Thank the referee for his suggestions. The fluorine atom is not formed in this reaction but achieved from the starting material, so it is not necessary to attach the F NMR spectra for the fluorine-containing compounds.

(5) The English grammar should be corrected for some expressions, for example, there are a little cumbersome in Introduction part, please carefully modify them.

■ Response: Thank the referee for his suggestion. We have checked and corrected this manuscript carefully. We have highlighted the changes with yellow background.

Overall, thanks again for your help in the evaluation of this manuscript. As described above, we revised this manuscript thoroughly and carefully. In addition, I believe the manuscript has been improved satisfactorily, and I hope you will find the revised manuscript is now suitable for the publication on *Nature Communications*.

Thank you in advance for your cooperation.

Reviewers' comments:

Reviewer #1 (Remarks to the Author):

The revised version of the paper is in my personal opinion lacking some important aspects. First of all, the assignment of the absolute configuration to the stereogenic axis is not correct. The authors report that compounds **5** have the M absolute configuration but this is wrong. They have reported a structure with a P absolute configuration. But, independently by the drawn structure, they do not reply correctly to my question. I wanted to know which is the relative configuration of the products **5**, because in this way, with the absolute configuration of the stereocenter, determined by X-ray of **3** series, they could easily know the absolute configuration of the chiral axis. Furthermore, the revision added on the SI, that is figure S16 wherein the absolute configuration of **5a** was obtained via DFT calculation, can not be accepted. They used these data to write in the manuscript that compounds **5** have a rotational barrier major than 30 kcal/mol. Where is coming from this data? From figure S16 I can see that DFT calculation estimates, I repeat estimates, the energy barrier for compound **5a** to be 27.8 and not 31.7 kcal/mol, as said in the text. You should explain the reason why compound **5a** decides to rotate following the highest syn-TS? Finally, this aspects highlight a scarce knowledge of the atropisomerism phenomena which can not go unnoticed and drive me to consider the rejection of the manuscript.

Dear Referees,

First, I hope you everyone is doing well as possible and remaining healthy under the circumstances. I am sorry that previous version was not sufficient because of the unexpected and serious coronavirus. Thank you very much for your positive comments regarding our above manuscript. We have read your comments carefully and made correction accordingly. In this revised version, we highlighted the changes made by giving the text a yellow background. Specifically, the following revisions were made:

Revisions made in reply to referee 1' comments:

Comments:

The revised version of the paper is in my personal opinion lacking some important aspects. First of all, the assignment of the absolute configuration to the stereogenic axis is not correct. The authors report that compounds 5 have the M absolute configuration but this is wrong. They have reported a structure with a P absolute configuration. But, independently by the drawn structure, they do not reply correctly to my question. I wanted to know which is the relative configuration of the products 5, because in this way, with the absolute configuration of the stereocenter, determined by X-ray of 3 series, they could easily know the absolute configuration of the chiral axis. Furthermore, the revision added on the SI, that is figure S16 wherein the absolute configuration of 5a was obtained via DFT calculation, can not be accepted. They used these data to write in the manuscript that compounds 5 have a rotational barrier major than 30 kcal/mol. Where is coming from this data? From figure S16 I can see that DFT calculation estimates, I repeat estimates, the energy barrier for compound 5a to be 27.8 and not 31.7 kcal/mol, as said in the text. You should explain the reason why compound 5a decides to rotate following the highest syn-TS? Finally, this aspects highlight a scarce knowledge of the atropisomerism phenomena which can not go unnoticed and drive me to consider the rejection of the manuscript.

■ Response: Thank the referee for his critical comments. In the past two weeks, we have repeated this work and carried out additional experiments for the stereochemistry for the assignment of the absolute configuration according to your good suggestions. The new

findings as following:

- (1) Fortunately, we have obtained the single crystal structure of product **5f** (see supplementary material, P69, and Figure 3 of Text), the X-ray analysis revealed that the absolute configuration of the axial chiral product is determined as (P, S). Therefore, on basis of the new experiment result, it is really difficult to judge the absolute configuration of atropisomeric compounds by circular dichroism because the conformation of the stereogenic axis is not easy to be fixed, causing a confused result of the theoretical calculation of the circular dichroism. Accordingly, this part of the calculation of circular dichroism is currently deleted. And now it was found that the stereochemistry of atropisomeric products was easily understand from the proposed reaction mechanism.

X-ray structures of **5f** (CCDC 1994220)

- (2) The energy barrier for representative compounds **3a**, **3ee**, **3ff**, **3gg**, **3hh**, **5a** was recalculated in the revised version (see Figure S10 of Supplementary Material P261). And accordingly, we have corrected the misassigned errors in the Text. Thank you very much for your careful check. For example, the energy barrier for compound **5a** is 44.2 kcal/mol that calculated by M06L method and 6-311G (d,p) basis set with Gaussian 09 program.

Figure S10. The relationship between the energy barriers of racemization and the bulkiness of substituents in *ortho*-position of the maleimides (Black line)

Overall, thanks again for your help in the evaluation of this manuscript. As described above, we revised this manuscript thoroughly and carefully. In addition, I believe the manuscript has been improved satisfactorily, and I hope you will find the revised manuscript is now suitable for the publication on Nature Communications.

REVIEWERS' COMMENTS:

Reviewer #3 (Remarks to the Author):

Hydrosilylation of unsaturated carbon-carbon bonds with hydrosilanes is one of the most important processes and can rank one of the most fundamental reactions in organosilicon chemistry. In this manuscript, Xu and co-workers reported a highly enantioselective Si-C coupling hydrosilylation of carbonyl-activated alkenes using palladium catalysis with excellent enantioselectivity and yields (up to 99% yield, >99:1 e.r.) for the synthesis of a wide of functionalized succinimides with or without axial C-N bond. And this process is quite useful because the Si-C coupling approach could be applied in in synthetic chemistry and photocatalysis for the enantioselective construction of unprecedentedly reported N-heterocycles bearing one to three carbon-stereogenic centres. These obtained results are particularly original and interesting. The authors also answered all the questions in a quite satisfactory manner. Now I recommend to publish this article in Nature Communications without any further modifications.